

# Effects of different proportions of stevia stalk on nutrient utilization and rumen fermentation in ruminal fluid derived from sheep

Xia Zhang[1], Ting Jiao[1], Shumin Ma[1], Xin Chen[1], Zhengwen Wang[2], Shengguo Zhao[2] and Yue Ren[3]

[1] College of Grassland Science, Gansu Agricultural University, Key Laboratory of Grassland Ecosystem, Gansu Agricultural University, Lanzhou, Gansu Province, China

[2] College of Animal Science and Technology, Gansu Agricultural University, Lanzhou, Gansu Province, China

[3] Institute of Animal Husbandry and Veterinary Medicine, Tibet Academy of Agricultural and Animal Husbandry Sciences, Lasa, Tibet Autonomous Region, China

## ABSTRACT

**Background**. Stevia straw is a byproduct of sugar crop stevia. It is a good feed material because of richness in nutrients and active substances (steviosides and flavonoids). However, due to improper utilization such as piling, burning and so on, it became a large amount of wasted straw resources and lead to environmental pollution.

**Methods**. We added 0%, 0.2%, 0.4%, 0.6%, 0.8%, 1.0% and 1.5% of stevia stalk to study the effects of different stevia stalk concentrations on nutrient utilization and rumen fermentation in sheep (based on sheep diet). *In vitro* fermentation method was used, with 17 repetitions for each treatment. All fermentation substrate based on sheep diet with different stevia stalk concentrations were fermented for 2 h, 6 h, 12 h, 24 h and 48 h, then the gas production, dry matter degradability (DMD), crude protein degradability (CPD), neutral detergent fiber degradability (NDFD), acid detergent fiber degradability (ADFD), pH, ammonia nitrogen ($NH_3$-N) and volatile fatty acids (VFAs) were determined.

**Results**. The results showed that at different fermentation time, the change trend of gas production in each teatment was basically same, but the maximum occurred in 1.0% treatment at 48 h. The DMD, CPD, NDFD and ADFD of sheep diets increased with fermentation time increasing, especially the $CPD_{48h}$, $NDFD_{48h}$ and $ADFD_{48h}$ of diets in 0.8%, 1.0% and 1.5% treatments were significantly higher than those in control ($P < 0.05$). The pH of fermentation substrate in each treatment remained within the normal range of 6.21~7.25. $NH_3$-$N_{24h–48h}$ in 0.8%, 1.0% and 1.5% treatments were higher than that in control. At 6 h–12 h, the total acid content of 0.8% and 1.0% treatments were significantly higher than those of other treatments ($P < 0.05$), it reached the highest in 1.0% treatment. According to overall evaluation, effect ranking of stevia stalk on sheep nutrient utilization was as follows: 1.0% >0.8% >1.5% >0.4% >0.6% >0.2%. Overall, 1.0% stevia stalk could promote nutrient degradation and sheep rumen fermentation.

Corresponding author
Ting Jiao, jiaot@gsau.edu.cn, jiaoting207@126.com

## INTRODUCTION

Stevia (*Stevia rebaudiana* Bert.) is a green plant used as a calorie-free sweetener due to the presence of stevioside in leaves (*Mehravaran et al., 2021*), which has become the third natural sugar source widely used in food, beverage, medical, daily chemical and other industries in the world (*Zhu, 2021*; *Sharma Saurabh et al., 2016*), in which stevia leaves was only used as raw materials. Thus, a large number of stevia waste residues and discarded stevia stalks were produced as by-products of the sugar industry. Among them, only a small part of residues were used as agricultural fertilizers (*Ding et al., 2016*), others were used for fuel combustion or even landfill, which resulted in a lot of waste of straw resources and environmental pollution. Stevia was also rich in amino acids, minerals and active ingredients such as steviol and polyphenols (chlorogenic acid, flavonoids) (*Xiong et al., 2022*; *Schiatti et al., 2022*). It became a research hotspot in the field of livestock and poultry feed as feed ingredients. Studied found that adding 0.3%∼0.6% stevia (whole plant) into feed for one month could improve dairy cow appetite (*Xie & Wang, 2010*). 5% stevia residue regulated poultry digestive function (*Shang, 2011*). Stevia or stevia glycosides could improve appetite in livestock (*Guo et al., 2016*; *Yu et al., 2020*; *Wang et al., 2011*; *Wang et al., 2014*), goats (*Han et al., 2019*), poultry (*Jiang et al., 2020*) and racehorses (*Ma & Ma, 2009*), improve pets anorexia and feed conversion rate and regulate intestinal microflora balance (*Ma & Ma, 2009*). In addition, *in vitro* rumen fermentation of stevia residue was a typical acetic acid type fermentation, which could promote rumen carbohydrate fermentation, improve energy utilization and VFAs production, reduce methane production during rumen fermentation, all of which could be conducive to reducing greenhouse gas emission (*He et al., 2017*). Stevia could also stimulate the biological metabolic activity of rumen microbe, promote ammonia nitrogen transferring to microbial protein, so accelerate microbial protein synthesis (*He et al., 2017*). Therefore, stevia can be used as a natural green feed material with various bioactive functions. However, current studies on stevia have focused on dairy cows and poultry, and few have been reported in sheep. In this study, the effects of stevia stalk with different proportions on nutrients utilization in sheep rumen fluid and rumen fermentation were studied by *in vitro* gas production method. The *in vitro* gas production method is a research method to simulate the dynamic process of feed degradation in rumen in the laboratory and then determine the rate of nutrient degradation in feeds (*Yang & Fang, 2015*; *Peñailillo et al., 2021*). Therefore, we hypothesized that there would be differences in *in vitro* gas production and substrate degradation rates, etc., through adding different stevia stalk proportions to sheep rations, and determined the effect of stevia stalk on sheep nutrient utilization and rumen fermentation by *in vitro* gas production method, to determine the key indexes such as gas production rate and substrate degradation rate, so as to screen the best adding ratio and provide a theoretical reference for scientific application of stevia stalk in sheep production.

**Table 1  Stevia stalk powder nutrients (air-dried basis %).**

| Material | DM(%) | CP | NDF | ADF | ESC | Ca | P | Ash |
|---|---|---|---|---|---|---|---|---|
| Stevia stalk | 96.64 | 3.76 | 57.62 | 40.04 | 10.90 | 0.73 | 0.13 | 5.34 |

Notes.

DM, dry matter; CP, crude protein; NDF, neutral detergent fiber; ADF, acid detergent fiber; ESC, monosaccharide; Ca, calcium; P, phosphorus; Ash, ash.

# MATERIALS & METHODS

## Test materials

Stevia stalk was collected from Wulan Town, Jingyuan County, Gansu Province, in November 2020. After drying at 65 °C for 48 h, initial moisture was determined, and plant material was crushed through a 40-mesh sieve for further use. Stevia samples were placed into nylon bags (9 cm × 5 cm, 400-mesh), which was placed into a 100 mL trachea with a chitin plug and plastic screw top. The basal diet was ground with a plant crusher and then prepared into dry matter substrate. The main nutrients in stevia stalk were shown in Table 1.

## Rumen fluid collection

All experimental protocols were approved by the Livestock Care Committee of Gansu Agricultural University (GAU-LC-2022-0555). Rumen fluid was obtained from animals at Zhonghua Sheep Farm in Lanzhou, and the experiment was conducted with the consent of the farmer. It was collected from three sheep (Small Tailed Han Sheep, 3 months old, male) at different sites in rumen, then mixed with $CO_2$ gas in an insulated bottle preheated to 39 °C. After the rumen fluid was extracted, the experimental sheep were in good health and had no adverse reaction. The bottle mouth was immediately covered, then transported to the laboratory quickly. All of the mixed contents were filtered through four layers of gauze and stored in containers in a 39 °C water bath. $CO_2$ gas was continuously injected, all of which were completed as soon as possible.

## Diet composition and nutrition

The trial sheep diet was made by Lanzhou Zhengda Co., Ltd. and formulated according to nutritional requirements of rams (body weight 45 kg, daily gain 50 g, fine to coarse ratio 4: 6) established by Agricultural Industry Standard of People's Republic of China (NY/T816-2004). Dietary composition and nutritional information was shown in Table 2.

## Test design

A single-factor experiment was conducted in seven treatments, these were, 0%, 0.2%, 0.4%, 0.6%, 0.8%, 1.0%, and 1.5% (dry matter basis) proportions of stevia stalk, added to total mixed basal diet and mixed evenly for fermenting substrate. The specific test design was shown in Table 3.

## *In vitro* fermentation method

Stevia stalk with concentrations of 0, 2, 4, 6, 8,10 and 15 g/kg was accurately weighed and placed in a homemade nylon bag (9 cm ×5 cm, 400-mesh) with 0.5000 g basal diet for substrate, numbered, sealed and then placed with forcep in the bottom of 100 mL tracheas,

**Table 2  Test diet composition and nutritional levels (dry basis).**

| Formula composition | Proportion/ % | Nutritional level | Content |
|---|---|---|---|
| Corn | 38 | DM / % | 86.0 |
| Corn germ meal | 20 | DE (MJ/kg) | 14.23 |
| Corn cob flour | 9 | ME (MJ/kg) | 11.67 |
| Rice Husk Powder | 8 | Ca (%) | 4.3 |
| Sprayed corn husk | 6 | P (%) | 1.9 |
| Corn husk | 5 | CP (%) | 9.4 |
| Cotton meal | 3 | | |
| Rapeseed meal | 2 | | |
| Soybean meal | 3.5 | | |
| Bean curd | 3.5 | | |
| 1% Premix additives | 1 | | |
| Salt | 1 | | |
| Total | 100 | | |

Notes.
Each kilogram of premix contains: vitamin A, 220000 IU, vitamin D, 372000 IU, vitamin E, 2000 IU, D-biotin ,40.0 mg, nicotinamide, 2000 mg, Mn, 710 mg, Zn, 2005 mg, Fe, 830.0 mg, Cu, 680.0 mg, Co, 12 mg.
DM, dry matter; DE, digestible energy; ME, metabolizable energy; Ca, calcium; P, phosphorus; CP, crude protein.

**Table 3  Test design.**

| Item | Control treatment | Test treatments | | | | | |
|---|---|---|---|---|---|---|---|
| Stevia stalk powder | 0% | 0.2% | 0.4% | 0.6% | 0.8% | 1.0% | 1.5% |
| Stevia stalk powder (% of DM in diet) concentration (g/kg) | 0 | 2 | 4 | 6 | 8 | 10 | 15 |

which were preheated to 39 °C for 30 min, in order to prevent gas from escaping, an appropriate amount of petroleum jelly was applied to the plunger of the syringe and then the filtered rumen fluid and artificial culture medium were mixed evenly at a volume ratio of 1: 2. 30 mL microbial culture mixture saturated with $CO_2$ was accurately measured and placed in each trachea, sealed with rubber tube and clips, and the initial volume (mL) of each trachea was recorded. Each treatment contained 17 replicates. In order to ensure the representativeness of the test sample, three blank samples were made when the sample was incubated to eliminate test errors. After reading the initial volume, the tracheas were immediately transferred to water bath for culture preheated to 39 °C (the water surface height of the bath should be higher than liquid surface height of trachea culture solution) for 48 h. All data of the gas production at 2 h, 6 h, 12 h, 24 h, 36 h and 48 h were recorded. After 2 h, 6 h, 12 h, 24 h and 48 h of fermentation, the fermentation tube was placed into an ice water bath to stop fermentation. Three replicates were removed after 2 h, 6 h, 12 h and 24 h fermenting respectively, keeping the last five replicates for 48 h fermenting were placed into an ice water bath to stop fermentation, collected fermentation broth and nylon bags for index measurements.

## Indexes and measurement methods

### Gas production

Gas production were recorded and the fermenting fluid were collected at 2 h, 6 h, 12 h, 24 h, 36 h, and 48 h; calculated cumulative gas production. The gas production for each time were calculated as fllows:

Gas production (mL/g) = (gas production in each tube recorded at each time - gas production in blank tubes at the same time)/fermentation substrate weight (*Yang et al., 2017*).

### Fermentation indexes

pH value of rumen fluid was determined by pH meter (P611 type). Determination of ammonia nitrogen concentration ($NH_3$-N), 10 mL of rumen fluid was centrifuged at 3,500 r/min for 10 min, 2 mL of supernatant was taken and placed in 15 mL test tube, then 8 mL of 0.2 mol/L hydrochloric acid was added to 10 mL, shaking, and the content of $NH_3$-N was determined by colorimetry (*Feng & Gao, 2010*).

For volatile fatty acids (VFAs), sample of each treatment at each time was centrifuged at 5,400 r/min for 10 min at 4 °C; 1 mL of supernatant was added to a 1.5 mL centrifuge tube containing 0.2 mL of 25% phosphonic acid solution containing internal standard 2-ethyl butyric acid. The samples were mixed well, placed in an ice water bath for more than 30 min, centrifuged at 10,000 r/min for 10 min, filtered through a 0.22 µL filter membrane, and detected by gas chromatography (*Li, 2021*).

### Rumen degradation rates

After 2 h, 6 h, 12 h, 24 h and 48 h of fermentation, nylon bag was washed with distilled water until clear. After natural drying, the nylon bag was transferred to an oven at 65 °C for 48 h and dried to a constant weight. The weight of the residue was weighed and used to determine DMD, CPD, ADFD and NDFD of different treatments at different times. ADF and NDF contents were determined by fiber washing method.

Every index of rumen degradation rate of feed was calculated as: (Mass of certain components in feed-the mass of associated components in residue)/the mass of associated components in feed ×100% (*Liu et al., 2021*).

### Data processing and statistical methods

Microsoft Excel 2010 was used for preliminary data collation. SPSS 23.0 software was used for single factor analysis of variance, duncan method was used for multiple comparisons. The results were expressed as means $\pm$ standard deviations, and $P < 0.05$ was considered significant. The data standardization of 48 h fermentation was transformed by membership function method.

The positive indexes were transformed according to the formula "Uij = (Xij-Ximin)/(Ximax-Ximin)"; the negative indexes were transformed according to the formula "Uij = 1 - (Xij-Ximin)/(Ximax-Ximin)". In the formula Uij was the index (j) membership function value of different stevia stalk treatments (i); Xij was the measured value, Ximax and Ximin were the maximum and minimum measured values, respectively. The membership value of each index was obtained by the membership function method, and the average
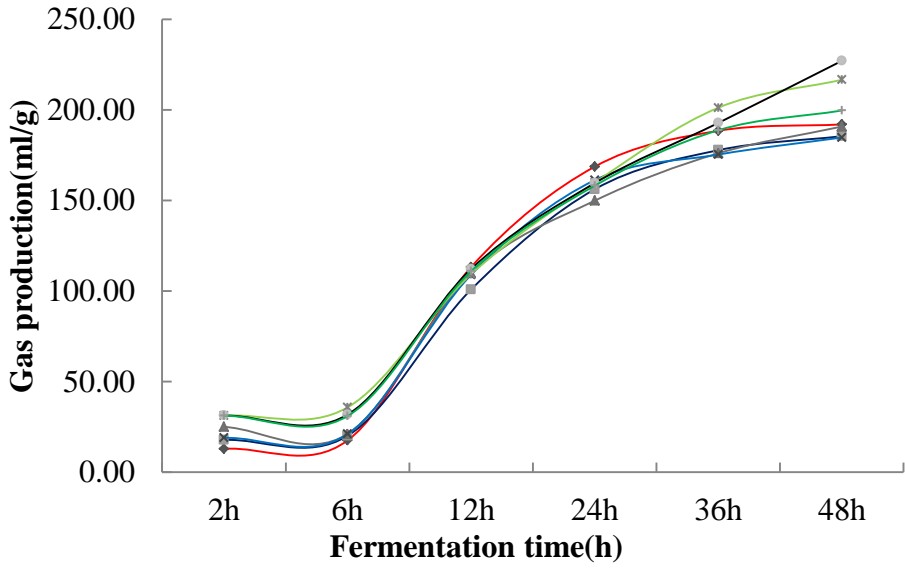

**Figure 1** **Variation curve of *in vitro* fermentation gas production under different stevia stalk treatments.** Different lines in the graph indicate different treatments, red represents 0% treatment, dark blue represents 0.2% treatment, dark gray represents 0.4% treatment, blue represents 0.6% treatment, light green represents 0.8% treatment, black represents 1.0% treatment, and dark green represents 1.5% treatment.

membership value of the measured index was calculated and used as the comprehensive evaluation of different stevia stalk treatments (*Liao et al., 2022*).

## RESULTS

### Effects of stevia stalk on gas production, DMD, CPD, NDFD and ADFD of sheep diet

#### *Effects of different stevia stalk proportions on gas production*

The change trend of fermented gas production remained the same under different stevia stalk treatments. It tended to be slight at 2 h-6 h and increased at 6 h-24 h in each treatment, but they were significantly higher in 0.8%, 1.0% and 1.5% treatment at 2 h-6 h than that in the control ($P < 0.05$). All of treatment reached to flat at 36 h-48 h among treatments. The maximum value was obtained in 0.8% treatment at 36 h ($P < 0.05$) while got the highest value in 1.0% treatment at 48 h ($P <$ 0.05) (Fig. 1; Table 4).

### Effects of stevia stalk on DMD, CPD, NDFD and ADFD

The DMD of sheep diet increased with fermentation time extention. It was lower in each treatment fermented at 2 h, 6 h and 12 h than that in the control. After 48 h of fermentation, there were no significant differences among treatments ($P > 0.536$). But the $DMD_{48h}$ in 0.2% and 1.0% treatments were higher than that in the control ($P > 0.05$). $CPD_{12 h}$, $CPD_{24h}$ and $CPD_{48h}$ in 1.0% and 1.5% treatments were higher than that in the control ($P < 0.05$). And the $ADFD_{48h}$ and $NDFD_{48h}$ of 0.8%, 1.0% and 1.5% treatments were significantly higher than those of other treatments ($P < 0.05$) (Table 5).

**Table 4  Significance among treatments.**

| Item | Fermentation time | | | | | |
|---|---|---|---|---|---|---|
| | **2 h** | **6 h** | **12 h** | **24 h** | **36 h** | **48 h** |
| 0% | c | d | a | a | ab | c |
| 0.2% | c | c | b | ab | b | c |
| 0.4% | ab | c | ab | b | b | c |
| 0.6% | bc | c | ab | ab | b | c |
| 0.8% | a | a | ab | ab | a | ab |
| 1.0% | a | b | a | ab | ab | a |
| 1.5% | a | b | a | ab | ab | bc |
| *P*- value | 0.000 | 0.000 | 0.148 | 0.218 | 0.072 | 0.004 |

**Notes.**
Different lowercase letters in the same column indicate significant differences ($P < 0.05$) between different stevia stalk concentrations at the same time.

## Effects of stevia stalk on *in vitro* fermentation characteristics

As shown in Table 6, the pH value of rumen fluid derived from sheep treated with different stevia concentrations fluctuated between 6.21 and 7.25. It increased first and then decreased in 0%, 0.2%, 0.4% and 0.6% treatments while decreased in 0.8%, 1.0% and 1.5% treatments during fermentation. In short, with the extension of fermentation, the pH of rumen fluid declined. At 48 h, the pH was significantly lower in the 0.8% and 1.0% treatments than in the control ($P < 0.05$).

The concentration of $NH_3$-N in sheep rumen fluid decreased with the extension of fermentation time. 0.4% treatment had significantly lower $NH_3$-$N_{48h}$ than other treatments ($P < 0.05$), while there had no significant different $NH_3$-$N_{48h}$ between other treatments and control ($P > 0.05$).

During period of 2 h-48 h, the concentrations of acetic acid and propionic acid varied significantly among treatments, while the concentrations of isobutyric acid, butyric acid, isovalerate and valerate changed only slightly and remained relatively stable. At 6 h and 12 h, the total acid content of 0.8% and 1.0% treatments were significantly higher than that of other treatments ($P < 0.05$), while at 48 h, it had no difference in 0.8%, 1.0% , and 1.5% treatment, compared to the control. But showed the higest value in 1.0% treatment.

## Comprehensive analysis

It was insufficient to evaluate the fermentation characteristics with single index. Using the membership function method to analyze the relevant indicators of rumen fermentation characteristics comprehensively could reflect the overall performance of rumen fermentation accurately. In this study, the membership function method was used to comprehensively analyze pH, $NH_3$-N, VFAs, DMD, CPD, NDFD and ADFD (Table 7). The higher the comprehensive analysis value, the better the fermentation of rumen nutrients. The results showed that order of the treatments was 1.0% >0.8% >1.5% >0.4% >0.6% >0.2%.
**Table 5  Effects of stevia stalk on DMD, CPD, NDFD and ADFD.**

| Items | | Fermentation time (h) | | | | |
|---|---|---|---|---|---|---|
| | | 2 h | 6 h | 12 h | 24 h | 48 h |
| DMD | 0% | 28.18 ± 1.14a | 28.08 ± 0.06a | 31.26 ± 0.85a | 32.92 ± 0.02a | 34.77 ± 0.78a |
| | 0.2% | 24.06 ± 0.36bc | 24.45 ± 0.22bc | 27.66 ± 0.25c | 29.83 ± 0.04b | 35.44 ± 0.27a |
| | 0.4% | 25.09 ± 0.92bc | 24.91 ± 0.4c | 28.34 ± 0.49bc | 31.19 ± 0.19ab | 34.96 ± 0.47a |
| | 0.6% | 25.59 ± 0.66b | 27.63 ± 0.26a | 29.67 ± 0.37abc | 32.26 ± 0.21a | 34.56 ± 0.51a |
| | 0.8% | 24.49 ± 0.42bc | 24.88 ± 0.21bc | 28.42 ± 0.03bc | 31.37 ± 0.33ab | 34.31 ± 0.79a |
| | 1.0% | 23.07 ± 0.17c | 25.19 ± 0.76bc | 27.64 ± 0.57c | 31.39 ± 1.13ab | 35.41 ± 0.29a |
| | 1.5% | 23.33 ± 0.41c | 25.91 ± 0.57b | 29.97 ± 1.36ab | 32.95 ± 0.77a | 34.19 ± 0.32a |
| P- value | | 0.002 | 0.000 | 0.018 | 0.013 | 0.536 |
| CPD | 0% | 0.42 ± 0.001f | 0.38 ± 0.01b | 0.44 ± 0.017bc | 0.51 ± 0.002bc | 0.62 ± 0.002b |
| | 0.2% | 0.48 ± 0.004b | 0.46 ± 0.001a | 0.47 ± 0.001ab | 0.51 ± 0.003bc | 0.69 ± 0.007ab |
| | 0.4% | 0.38 ± 0.001 g | 0.4 ± 0.007b | 0.41 ± 0.014c | 0.52 ± 0.009bc | 0.63 ± 0.002b |
| | 0.6% | 0.43 ± 0.002e | 0.46 ± 0a | 0.45 ± 0.012abc | 0.5 ± 0.008c | 0.62 ± 0.004b |
| | 0.8% | 0.44 ± 0.002d | 0.39 ± 0.003b | 0.49 ± 0.002ab | 0.58 ± 0.006b | 0.73 ± 0.015a |
| | 1.0% | 0.51 ± 0.002a | 0.44 ± 0.023a | 0.5 ± 0.016a | 0.66 ± 0.011a | 0.77 ± 0.053a |
| | 1.5% | 0.46 ± 0.001c | 0.44 ± 0.016a | 0.5 ± 0.028a | 0.67 ± 0.061a | 0.77 ± 0.045a |
| P- value | | 0.000 | 0.001 | 0.004 | 0.000 | 0.002 |
| NDFD | 0% | 38.8 ± 0.96a | 41.9 ± 0.81cd | 63.23 ± 0.67c | 63.18 ± 0.96bc | 58.03 ± 0.57b |
| | 0.2% | 31.93 ± 0.85bc | 38.5 ± 0.6cd | 58.38 ± 1.44d | 57.83 ± 0.82cd | 52.44 ± 1.85c |
| | 0.4% | 39.3 ± 1.21a | 36.28 ± 0.4d | 69.51 ± 0.56b | 60.94 ± 0.55cd | 55.93 ± 0.65c |
| | 0.6% | 30.1 ± 0.07c | 42.79 ± 1.09ab | 71.07 ± 1.06ab | 69.75 ± 0.24a | 58.25 ± 0.95b |
| | 0.8% | 34.23 ± 0.29b | 46.19 ± 0.44a | 66.18 ± 0.97c | 54.8 ± 2.79d | 71.13 ± 2.11a |
| | 1.0% | 24.44 ± 1.36d | 36.37 ± 2.11d | 63.05 ± 1.82c | 60.07 ± 0.94cd | 73.18 ± 1.01a |
| | 1.5% | 31.31 ± 0.5c | 46.34 ± 1.41a | 73.12 ± 0.43a | 68.6 ± 3.9ab | 72.4 ± 0.94a |
| P- value | | 0.000 | 0.000 | 0.000 | 0.001 | 0.000 |
| ADFD | 0% | 26.52 ± 1.05bc | 36.95 ± 0.15d | 57.7 ± 0.42d | 56.61 ± 0.34e | 48.37 ± 0.49d |
| | 0.2% | 28.62 ± 0.49b | 36.37 ± 0.34d | 58.17 ± 0.07d | 70.11 ± 0.22ab | 45.73 ± 0.52e |
| | 0.4% | 22.53 ± 0.83de | 36.37 ± 0.2d | 60.99 ± 0.19c | 59.79 ± 0.93d | 45.45 ± 0.36e |
| | 0.6% | 24.33 ± 0.69cd | 36.18 ± 0.61d | 66.07 ± 1.01b | 57.35 ± 1.6de | 59.19 ± 0.44b |
| | 0.8% | 33.44 ± 0.32a | 42.95 ± 0.21b | 68.2 ± 0.18a | 66.72 ± 1.34c | 67.34 ± 0.71a |
| | 1.0% | 21.22 ± 1.42e | 38.9 ± 0.79c | 65.35 ± 0.76b | 72.28 ± 0.99a | 52.76 ± 1.09c |
| | 1.5% | 28.46 ± 0.42b | 48.66 ± 0.31a | 67.17 ± 0.86ab | 68.58 ± 0.65bc | 66.48 ± 1.35a |
| P- value | | 0.000 | 0.000 | 0.000 | 0.000 | 0.000 |

**Notes.**

Different lowercase letters in same column indicate significant differences ($P < 0.05$) between different stevia stalk concentrations at the same time and the same index; this also applies to the table below.

DMD, dry matter degradability; CPD, crude protein degradability; NDFD, neutral detergent fiber degradability; ADFD, acid detergent fiber degradability.

# DISCUSSION

## Substrate degradation rate

Dry matter degradation rate, gas production and *in vivo* digestibility are highly correlated (*Blümmel, Makkar & Becker, 1997*). In this experiment, the trend of gas production with different concentrations was consistent, and they were stable at 2 h-6 h, it was because the rumen microbial activity was weak and a small amount of gas produced in early
**Table 6 Changes of pH, NH₃-N and VFAs in rumen fluid in different Stevia stalk treatments (mmol L⁻¹).**

| Items | | Fermentation time (h) | | | | |
|---|---|---|---|---|---|---|
| | | 2 h | 6 h | 12 h | 24 h | 48 h |
| pH | 0% | 6.98 ± 0.01b | 7.25 ± 0.01a | 6.85 ± 0.01a | 6.63 ± 0.09b | 6.48 ± 0.14bc |
| | 0.2% | 6.93 ± 0.01b | 7.24 ± 0.01a | 6.88 ± 0.01a | 6.83 ± 0.01a | 6.55 ± 0.1ab |
| | 0.4% | 6.93 ± 0.02b | 7.21 ± 0.02ab | 6.87 ± 0.01a | 6.84 ± 0.01a | 6.73 ± 0.01a |
| | 0.6% | 7.11 ± 0.1a | 7.19 ± 0.01b | 6.88 ± 0.02a | 6.8 ± 0.03a | 6.68 ± 0.07ab |
| | 0.8% | 6.9 ± 0.01b | 6.83 ± 0.01c | 6.57 ± 0.01c | 6.6 ± 0.02b | 6.24 ± 0.02d |
| | 1.0% | 6.89 ± 0.01b | 6.81 ± 0.02c | 6.62 ± 0b | 6.56 ± 0.02b | 6.21 ± 0.04d |
| | 1.5% | 6.92 ± 0.01b | 6.83 ± 0.01c | 6.6 ± 0b | 6.52 ± 0.01b | 6.29 ± 0.02cd |
| *P*- value | | 0.014 | 0.000 | 0.000 | 0.000 | 0.000 |
| NH₃-N | 0% | 20.08 ± 0.28ab | 16.21 ± 1.98c | 12.17 ± 0.33e | 26.04 ± 0.24c | 37.69 ± 0.87ab |
| | 0.2% | 20.93 ± 1.56ab | 19.37 ± 0.51bc | 17.53 ± 0.66d | 23.93 ± 0.34de | 37.27 ± 3.75ab |
| | 0.4% | 16.7 ± 1.26b | 17.3 ± 2.64c | 14.2 ± 1.37e | 22.1 ± 0.57e | 32.92 ± 1.16c |
| | 0.6% | 17.97 ± 1.99ab | 15.44 ± 2.42c | 20.7 ± 0.08c | 24.88 ± 0.85cd | 34.54 ± 1.17ab |
| | 0.8% | 18.94 ± 0.71ab | 27.17 ± 1.96a | 31.41 ± 0.56b | 44.15 ± 0.72ab | 38.67 ± 0.43b |
| | 1.0% | 22.48 ± 0.55a | 24.53 ± 3.48ab | 35.97 ± 1.07a | 45.92 ± 0.3a | 43.39 ± 1.91a |
| | 1.5% | 20.05 ± 3.15ab | 27.85 ± 1.57b | 30.48 ± 0.48b | 42.39 ± 0.85b | 43.67 ± 1.49a |
| *P*- value | | 0.294 | 0.004 | 0.000 | 0.000 | 0.000 |
| Acetic acid | 0% | 12.27 ± 0.06c | 18.2 ± 0.1d | 18.81 ± 0.04d | 16.69 ± 1.36d | 24.51 ± 1.79ab |
| | 0.2% | 12.02 ± 0.52c | 13.48 ± 0.14e | 17.86 ± 0.14d | 22.12 ± 1.03bc | 13.89 ± 3.52c |
| | 0.4% | 16.22 ± 0.1b | 13.67 ± 0.34e | 18.52 ± 0.02d | 23.51 ± 1.04bc | 24.36 ± 4.07ab |
| | 0.6% | 11.42 ± 0.51c | 12.02 ± 0.3f | 20.07 ± 0.04c | 28.52 ± 0.9a | 16.88 ± 0.99bc |
| | 0.8% | 15.71 ± 0.03b | 23.46 ± 0.56b | 29.8 ± 0.64a | 20.77 ± 0.43c | 27.49 ± 0.74a |
| | 1.0% | 18.1 ± 0.17a | 24.62 ± 0.31a | 26.27 ± 0.43b | 24.44 ± 0.57b | 31.61 ± 1.17a |
| | 1.5% | 15.41 ± 0.14b | 19.74 ± 0.25c | 20.3 ± 0.56c | 21.83 ± 0.41bc | 22.27 ± 2.08abc |
| *P*- value | | 0.000 | 0.000 | 0.000 | 0.000 | 0.006 |
| Propionic acid | 0% | 6.15 ± 0.13d | 5.22 ± 0.01b | 7.28 ± 0.21cd | 5.17 ± 0.02d | 15.74 ± 4.61bc |
| | 0.2% | 5.19 ± 0.11d | 6.78 ± 0.23b | 8.61 ± 0.11c | 5.22 ± 0.01d | 8.77 ± 1.74d |
| | 0.4% | 13.77 ± 0.25a | 5.18 ± 0.01b | 9.38 ± 0.03c | 5.24 ± 0.01d | 10.44 ± 0.88cd |
| | 0.6% | 5.56 ± 0.41d | 5.19 ± 0.01b | 5.13 ± 0.01d | 11.62 ± 1.06c | 8.69 ± 1.01d |
| | 0.8% | 8.46 ± 0.48c | 19.46 ± 2.36a | 18.64 ± 1.8b | 17.92 ± 0.16b | 20.04 ± 0.07ab |
| | 1.0% | 11.78 ± 0.76b | 5.54 ± 0.21b | 24.2 ± 0.45a | 20.42 ± 0.67a | 22.45 ± 0.63a |
| | 1.5% | 9.34 ± 0.66c | 5.31 ± 0.1b | 16.66 ± 1.23b | 17.99 ± 0.3b | 18.96 ± 1.55ab |
| *P*- value | | 0.000 | 0.000 | 0.000 | 0.000 | 0.000 |
| Isobutyric acid | 0% | 0.48 ± 0.01bc | 0.41 ± 0d | 0.54 ± 0.01abc | 0.69 ± 0.02ab | 0.78 ± 0.13abc |
| | 0.2% | 0.48 ± 0.01bc | 0.5 ± 0.01c | 0.52 ± 0.01cd | 0.47 ± 0.04e | 0.56 ± 0.03c |
| | 0.4% | 0.5 ± 0ab | 0.48 ± 0c | 0.55 ± 0ab | 0.57 ± 0d | 0.6 ± 0.06bc |
| | 0.6% | 0.48 ± 0c | 0.49 ± 0c | 0.48 ± 0.04d | 0.61 ± 0.02cd | 0.63 ± 0.01abc |
| | 0.8% | 0.5 ± 0abc | 0.59 ± 0.02a | 0.61 ± 0.01b | 0.72 ± 0a | 0.84 ± 0ab |
| | 1.0% | 0.52 ± 0.02a | 0.56 ± 0.01ab | 0.68 ± 0.04a | 0.72 ± 0.01a | 0.86 ± 0.02a |
| | 1.5% | 0.5 ± 0.01abc | 0.55 ± 0.01b | 0.54 ± 0abc | 0.65 ± 0bc | 0.84 ± 0.03ab |

**Table 6** (*continued*)

| Items | | 2 h | 6 h | 12 h | 24 h | 48 h |
|---|---|---|---|---|---|---|
| | | | | **Fermentation time (h)** | | |
| *P*- value | | 0.011 | 0.000 | 0.000 | 0.000 | 0.033 |
| | 0% | 1.94 ± 0.01b | 2.19 ± 0.02c | 3.02 ± 0.07de | 3.48 ± 0.05e | 4.67 ± 0.07ab |
| | 0.2% | 2.43 ± 0.23b | 2.05 ± 0.03c | 2.84 ± 0.05e | 3.96 ± 0.04d | 2.66 ± 0.62c |
| | 0.4% | 3.24 ± 0.05a | 2 ± 0.01c | 3.12 ± 0.06d | 4.26 ± 0.02cd | 4.37 ± 0.23ab |
| Butyric acid | 0.6% | 2.13 ± 0.32b | 2.03 ± 0.02c | 3 ± 0.1de | 4.7 ± 0.23a | 3.98 ± 0.22b |
| | 0.8% | 3.07 ± 0.04a | 4.03 ± 0.32a | 4.59 ± 0.04b | 4.35 ± 0bc | 5.06 ± 0.11ab |
| | 1.0% | 3.4 ± 0.24a | 3.71 ± 0.07ab | 5.44 ± 0a | 4.59 ± 0.07ab | 5.32 ± 0.05a |
| | 1.5% | 3.11 ± 0.06a | 3.47 ± 0.14b | 3.44 ± 0.05c | 4.19 ± 0.09cd | 4.86 ± 0.27ab |
| *P*- value | | 0.000 | 0.000 | 0.000 | 0.000 | 0.000 |
| | 0% | 0.53 ± 0.01ab | 0.52 ± 0b | 0.56 ± 0.01cd | 0.88 ± 0.14a | 1.05 ± 0.16b |
| | 0.2% | 0.55 ± 0.02a | 0.48 ± 0.02b | 0.53 ± 0.01d | 0.63 ± 0b | 0.62 ± 0.12c |
| | 0.4% | 0.54 ± 0ab | 0.49 ± 0.01b | 0.57 ± 0.01cd | 0.56 ± 0.04b | 0.91 ± 0.05bc |
| Isovaleric acid | 0.6% | 0.49 ± 0.01bc | 0.5 ± 0b | 0.53 ± 0.01d | 0.69 ± 0.03b | 0.87 ± 0.09bc |
| | 0.8% | 0.52 ± 0ab | 0.62 ± 0.02a | 0.76 ± 0.02b | 1.03 ± 0.03a | 1.83 ± 0.14a |
| | 1.0% | 0.53 ± 0.01ab | 0.59 ± 0.01a | 0.83 ± 0.02a | 1.03 ± 0a | 1.6 ± 0.09a |
| | 1.5% | 0.45 ± 0.04c | 0.58 ± 0.01a | 0.58 ± 0.01c | 0.9 ± 0.03a | 1.46 ± 0.03a |
| *P*- value | | 0.023 | 0.000 | 0.000 | 0.000 | 0.000 |
| | 0% | 0.56 ± 0b | 0.66 ± 0.01c | 1.02 ± 0.02d | 1.74 ± 0.29a | 2.51 ± 0.33a |
| | 0.2% | 0.71 ± 0.12b | 0.63 ± 0.01c | 0.94 ± 0.01d | 1.36 ± 0.02b | 1.08 ± 0.35c |
| | 0.4% | 1.01 ± 0.01a | 0.63 ± 0c | 1.05 ± 0d | 1.54 ± 0ab | 1.62 ± 0.06bc |
| Valeric acid | 0.6% | 0.64 ± 0.12b | 0.62 ± 0.01c | 0.92 ± 0.01d | 1.77 ± 0.03a | 1.47 ± 0.17bc |
| | 0.8% | 0.97 ± 0.01a | 1.33 ± 0.09a | 1.53 ± 0.02b | 1.67 ± 0.01ab | 2.09 ± 0.05ab |
| | 1.0% | 1.05 ± 0.06a | 1.25 ± 0.01ab | 1.72 ± 0.11a | 1.77 ± 0.01a | 2.14 ± 0.04ab |
| | 1.5% | 0.99 ± 0.01a | 1.18 ± 0.03b | 1.22 ± 0.01c | 1.54 ± 0ab | 1.91 ± 0.07ab |
| *P*- value | | 0.000 | 0.000 | 0.000 | 0.145 | 0.005 |
| | 0% | 21.92 ± 0.04c | 27.18 ± 0.12cd | 31.23 ± 0.26de | 28.64 ± 1.87d | 49.05 ± 3.28ab |
| | 0.2% | 21.38 ± 0.8cd | 23.92 ± 0.33de | 31.3 ± 0.3de | 33.75 ± 1.07c | 33.36 ± 8.44c |
| | 0.4% | 35.28 ± 0.4a | 22.45 ± 0.34e | 33.19 ± 0.11d | 35.69 ± 1.03c | 42.31 ± 5.22bc |
| Total acid | 0.6% | 19.78 ± 0.58d | 20.84 ± 0.29e | 30.13 ± 0.1e | 47.91 ± 1.87b | 32.51 ± 3.2c |
| | 0.8% | 29.22 ± 0.42b | 49.49 ± 3.36a | 55.93 ± 1.2b | 46.45 ± 0.37b | 57.79 ± 0.65ab |
| | 1.0% | 35.38 ± 0.46a | 36.27 ± 0.05b | 59.14 ± 1.05a | 52.96 ± 1.33a | 63.62 ± 1.72a |
| | 1.5% | 29.79 ± 0.91b | 30.83 ± 0.46c | 42.73 ± 1.59c | 47.1 ± 0.8b | 50.3 ± 3.69ab |
| *P*- value | | 0.000 | 0.000 | 0.000 | 0.000 | 0.003 |

**Notes.**

NH$_3$-N, ammoniacal nitrogenin; VFAs, volatile fatty acids.

**Table 7  Comprehensive indexes, weights, membership function values and rankings.**

| | Item | 0.2% | 0.4% | 0.6% | 0.8% | 1.0% | 1.5% |
|---|---|---|---|---|---|---|---|
| Weights | pH | 0.0542 | 0.0645 | 0.0615 | 0.0898 | 0.0909 | 0.0854 |
| | $NH_3$-N | 0.0554 | 0.0767 | 0.0692 | 0.0843 | 0.0756 | 0.0715 |
| | Acetic acid | 0.0967 | 0.0674 | 0.0920 | 0.0771 | 0.0675 | 0.0912 |
| | Propionic acid | 0.0984 | 0.1010 | 0.1148 | 0.0679 | 0.0610 | 0.0688 |
| | Isobutyric acid | 0.0763 | 0.0871 | 0.0785 | 0.0803 | 0.0790 | 0.0769 |
| | Butyric acid | 0.0962 | 0.0716 | 0.0744 | 0.0798 | 0.0764 | 0.0796 |
| | Isovaleric acid | 0.0928 | 0.0773 | 0.0765 | 0.0496 | 0.0571 | 0.0596 |
| | Valeric acid | 0.1274 | 0.1038 | 0.1082 | 0.1039 | 0.1021 | 0.1083 |
| | Total acid | 0.0806 | 0.0776 | 0.0956 | 0.0740 | 0.0671 | 0.0808 |
| | DMD | 0.0538 | 0.0666 | 0.0638 | 0.0869 | 0.0861 | 0.0842 |
| | CPD | 0.0496 | 0.0659 | 0.0634 | 0.0737 | 0.0705 | 0.0670 |
| | NDFD | 0.0606 | 0.0695 | 0.0503 | 0.0706 | 0.0867 | 0.0664 |
| | ADFD | 0.0580 | 0.0713 | 0.0518 | 0.0621 | 0.0798 | 0.0603 |
| Membership function value | pH | 0.6538 | 1.0000 | 0.9038 | 0.0577 | 0.0000 | 0.1538 |
| | $NH_3$-N | 0.4047 | 0.0000 | 0.1507 | 0.5349 | 0.9740 | 1.0000 |
| | Acetic acid | 0.0000 | 0.5909 | 0.1687 | 0.7675 | 1.0000 | 0.4729 |
| | Propionic acid | 0.0058 | 0.1272 | 0.0000 | 0.8249 | 1.0000 | 0.7464 |
| | Isobutyric acid | 0.0000 | 0.1333 | 0.2333 | 0.9333 | 1.0000 | 0.9333 |
| | Butyric acid | 0.0000 | 0.6429 | 0.4962 | 0.9023 | 1.0000 | 0.8271 |
| | Isovaleric acid | 0.0000 | 0.2397 | 0.2066 | 1.0000 | 0.8099 | 0.6942 |
| | Valeric acid | 0.0000 | 0.3776 | 0.2727 | 0.7063 | 0.7413 | 0.5874 |
| | Total acid | 0.0270 | 0.3148 | 0.0000 | 0.7974 | 1.0000 | 0.5714 |
| | DMD | 1.0000 | 0.6130 | 0.2965 | 0.3322 | 0.7904 | 0.0000 |
| | CPD | 0.4444 | 0.0689 | 0.0000 | 0.7334 | 0.9889 | 1.0000 |
| | NDFD | 0.0000 | 0.1681 | 1.0000 | 0.9013 | 0.2803 | 0.9622 |
| | ADFD | 0.0127 | 0.0000 | 0.6273 | 1.0000 | 0.3338 | 0.9604 |
| Comprehensive evaluation value | | 0.1372 | 0.3139 | 0.2838 | 0.7041 | 0.7355 | 0.6559 |
| Total ranking | | 6 | 4 | 5 | 2 | 1 | 3 |

fermentation stage, while to 6 h-36 h, rumen microbe proliferated, acting on substrate completely to be digested enough, thus producing large amounts of gas rapidly. The gas production decreased slowly with the nutrients being consumed. *Menke et al. (1979)* showed that the rate of forage organic matter degradation was positively correlated with gas production during *in vitro* incubation, *i.e.,* higher gas production indicated higher forage degradation in rumen and higher microbial activity. The results of this study showed that the $CPD_{48h}$, $NDFD_{48h}$ and $ADFD_{48h}$ of 0.6%, 0.8%, 1.0% and 1.5% stevia stalk treatments were higher than those without stevia stalk, and 1.0% treatment has the highest gas production at 48 h, which was consistent with the result of $DMD_{48h}$, $CPD_{48h}$ and $NDFD_{48h}$. It showed that appropriate concentration of stevia stalk could improve sheep rumen digestive function and promote crude fiber digestion and absorption in diet, adding 1.0% stevia stalk could promote the microbe fermentation activities in rumen more effectively, thus improved feed digestibility. Studies showed that the active ingredients such as stevia or flavonoids in

*stevia rebaudiana* could reduce rumen protozoa and increase cellulolytic bacteria amount to facilitate fiber decomposition and change rumen micro ecological environment (*He et al., 2017*). This was consistent with the results of the Sarnataro's study (*2020*). Studies reported that sea buckthorn flavonoids (*Bai et al., 2020*) and sea onion flavonoids (*Bao et al., 2015*) significantly increased rumen gas production, which was consistent with the results of the present experimental study. Its specific mechanism needs further research in feeding experiments.

## Rumen fermentation characteristics

The pH value is a comprehensive index reflecting rumen fermentation level, it not only reflects the strength of rumen microbial activity but also serves as an important indicator to evaluate nutrients decomposition status and assess rumen internal environment stability. The normal rumen pH is 5.5 ~7.5, and pH values higher or lower than this range will affect normal fermentation in rumen. When pH value is lower than 6.0, the number of fiber-decomposing bacteria and protozoa will decrease, and the cellulose and protein degradation rates in diet will also decrease (*Liu et al., 2012*; *Krause & Oetzel, 2005*). In this experiment, the pH range was 6.21 to 7.25, and high stevia stalk concentration (0.8%, 1.0%, 1.5%) could decrease the pH of the rumen culture medium, but the final pH still remained within the normal physiological range. This was consistent with the results of He's study (*2017*). It indicated that the artificial rumen was in a normal fermentation state and had not adverse effects on normal metabolism of rumen microbe, ensuring rumen microbe normal growth and reproduction. An explanation was that rumen microbe decomposed starch and structural carbohydrates in feed, producing a large number of VFAs. Moreover, with the extension of fermentation, VFAs was accumulated in fermentation tubes, resulting in a further decline in pH value. Consistent with the result of a negative correlation between VFAs and rumen pH by *Stritzler et al. (1998)*.

The $NH_3$-N concentration in rumen is an important indicator of nitrogen metabolism, microbial protein synthesis and protein degradation. Maintaining an appropriate $NH_3$-N concentration is a prerequisite for microbial protein synthesis in rumen (*Feng et al., 2017*). A suitable rumen $NH_3$-N concentration is beneficial to growth of microbe, and higher $NH_3$-N concentration is beneficial to fibrolytic bacteria growth and reproduction and DM degradation. The most suitable $NH_3$-N concentration range for microbial activity was 0.35 ~29 mg/dL (*Galdi et al., 2002*). In this experiment, the $NH_3$-N concentration was 12.17 ~45.92 mg/dL, and $NH_3$-N concentrations were relatively high in all treatments for 48 h. *He et al. (2017)* found that stevia residue reduced ruminal $NH_3$-N concentration, contrary to the results of the present study. This might be related to factors such as feed type, feeding form and *in vitro* fermentation. In this experiment, after 12 h of rumen fermentation, $NH_3$-N concentration in each treatment increased significantly, which might be due to the longer retention time of chyme in rumen and the longer interaction time between rumen microbe and feed particles (*He et al., 2020*). This might also be associated with lower microbial utilization. With the extension of fermentation, the pH of rumen fluid decreased, and the decreased pH environment was not conducive to the growth of microbe that decomposed structural carbohydrates. The utilization efficiency of $NH_3$-N by microbe

that decompose structural carbohydrates decreased, resulting in increasing of $NH_3$-N concentration in rumen. This was also consistent with the decreasing trends of the pH value.

VFAs in the rumen are important indicators of rumen fermentation in sheep and can reflect the environmental conditions in the rumen, the fermentation degree and the type of feed in rumen (*Liu et al., 2012*). In this experiment, at 48 h of fermentation, the content of acetic acid and propionic acid increased in all treatments, and acetic acid contents were higher than propionic acid contents, and the total acid content of 1.0% treatment was the highest. It could be seen that adding 1.0% stevia stalk could increase the concentration of VFA *in vitro* fermentation, better improve rumen fermentation capacity, maintain the normal growth, development and reproduction of rumen microbe and ensure the normal function of rumen, which was consistent with the results of pH value decreased to the lowest and $NH_3$-N concentration increased. However, *Jiang (2019)* found that the addition of stevia pellets to lactating cows did not affect rumen fermentation. Studies had shown that *in vitro* rumen fermentation of stevia residue was a typical acetic acid type fermentation, which could promote rumen carbohydrate fermentation, improve energy utilization and VFAs production (*He et al., 2017*). This indicated that stevia stalk also had good feeding value as a by-product. Stevia extract could reduce the number of rumen protozoa, thereby affecting rumen metabolism (*Chiara & Mauro, 2020*). It was also reported that stevia could help digest and prevent constipation. Stevia could promote the proliferation of bifidobacteria and lactobacillus in human body, improve human immunity, prevent and treat constipation and diarrhea (*Xu, 2013*). *Stevia rebaudiana* residue could significantly inhibit the growth of harmful bacteria and promote the proliferation of beneficial bacteria (*Zhong et al., 2022*). Therefore, it was speculated that the glycosides in stevia stalk also had an effect on sheep gastrointestinal fermentation. However, whether the increase of VFAs content in this experiment was related to steviosides in *stevia rebaudiana*, its mechanism needs further study.

## CONCLUSIONS

Evaluating the nutritional value of ruminant feed by *in vitro* gas production was feasible, and adding stevia to the diet could be used by ruminants and could improve feed conversion, promote rumen fermentation and, to a certain extent, reduce breeding costs. Under the experimental conditions, the appropriate level of stevia stalk supplementation was 1.0%.

## ACKNOWLEDGEMENTS

We thank Chen Zheng for assistance in the laboratory.

### Funding

This work has been supported by Gansu University Innovation Fund (No. 2022B-096) in 2022, Major science and technology projects in Tibet Autonomous Region

(XZ202101ZD0001N), South Xinjiang Key Industry Innovation Development Support Program Project (No. 2022DB017). The funders had no role in study design, data collection and analysis, decision to publish, or preparation of the manuscript.

### Grant Disclosures

The following grant information was disclosed by the authors:
Gansu University Innovation Fund: 2022B-096.
Tibet Autonomous Region: XZ202101ZD0001N.
South Xinjiang Key Industry Innovation Development Support Program Project: 2022DB017.

### Competing Interests

The authors declare there are no competing interests.

### Author Contributions

- Xia Zhang conceived and designed the experiments, performed the experiments, analyzed the data, prepared figures and/or tables, and approved the final draft.
- Ting Jiao conceived and designed the experiments, authored or reviewed drafts of the article, and approved the final draft.
- Shumin Ma performed the experiments, prepared figures and/or tables, and approved the final draft.
- Xin Chen performed the experiments, prepared figures and/or tables, and approved the final draft.
- Zhengwen Wang performed the experiments, prepared figures and/or tables, and approved the final draft.
- Shengguo Zhao conceived and designed the experiments, authored or reviewed drafts of the article, and approved the final draft.
- Yue Ren analyzed the data, prepared figures and/or tables, and approved the final draft.

### Animal Ethics

The following information was supplied relating to ethical approvals (i.e., approving body and any reference numbers):

Application Form for Ethical Review of Experimental Animal Welfare of Gansu Agricultural University.

### Data Availability

The raw measurements are available in the Supplementary File.

### Supplemental Information

Supplemental information for this article can be found online at http://dx.doi.org/10.7717/peerj.14689#supplemental-information.

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
