# Peer review of "Effects of different proportions of stevia stalk on nutrient utilization and rumen fermentation in ruminal fluid derived from sheep"

_PeerJ, doi:10.7717/peerj.14689_

## Round 0.1 · original submission · Major Revisions

Please address the concerns of the reviewers and revise the manuscript accordingly.

Reviewer 1 ·

Basic reporting

PeerJ - Manuscript 77911 submitted and entitled “Effects of different proportions of Stevia stalk on nutrient utilization and rumen fermentation in ruminal fluid derived from sheep” studied the potential of adding Stevia stalk to sheep diets on nutrient utilization and rumen fermentation were in studied in this experiment, which will provide a theoretical reference for scientific application of Stevia stalk in sheep production to improve production performance and achieve green ecological feed products using In vitro fermentation method. The authors concluded that adding Stevia to the diet might be beneficial to ruminants and evaluating the nutritional value of ruminant feed. I have some major comments that might be considered:
 English wording needs to be checked in this section. Please also check your text, as some information are repeated.
 The hypothesis is not clear, and objectives in introduction does not completely fit to each other. I would add a few sentences explaining what additional information your study will bring.
 Discussion needs to be rewritten with emphasis on important and economical findings
 There are no references reflecting 2022, it should be for references updating.
 The discussion should include further arguments between studies or results found in the literature.
 Overall, I recommend a detailed re-reading to rephrase those sentences. I recommend a major revision.

Experimental design

The major comment is about improving all manuscript section. It needs to be better structured and detailed.
 Throughout the manuscript, several scientific names and Latin expressions should be italicized.

Validity of the findings

In the introduction section, please emphasis on the added value/novelty of this article as there are some researches similar in the literature.

Reviewer 2 ·

Basic reporting

The author and their team have done a comprehensive analysis of the stevia stalk proportions derived from sheep. The articles cited almost cover all the work that has been done in this field of research. The manuscript describes the goal of their research in a detailed fashion and have addressed that almost all work in this field has been done on livestock and poultry. Therefore, understanding the fermentation levels in sheep is a critical area which has been addressed here.

Experimental design

very well defined experimental design and thorough studies have been conducted to achieve the results presented in this manuscript.

Validity of the findings

1. Very well defined methods, experimental details and subsequent analysis has been conducted.
2. The data presented is statistically significant and can be reproduced by others in the field with the information provided.
3. the conclusions drawn from the findings correlates with the hypothesis and 1.0% stevia stalk concentration at 48h has been accurately determined to provide the highest gas emissions.
4. pH is a critical factor in gut microbial studies and authors have good data to backup their findings.
5. the conclusions are well stated and circle back with initial hypothesis.

Additional comments

no comments

---

## Round 0.2 · accepted · Accept

Since the reviewer of the original submission did not respond to my invitation and since time is running, I decided to make a decision based on my own reading of the revised manuscript and response to the reviewers. In my view, all issues pointed by the reviewers are adequately addressed and the revised manuscript is acceptable now.